# Discrimination of Object Curvature Based on a Sparse Tactile Sensor Array

**DOI:** 10.3390/mi11060583

**Published:** 2020-06-10

**Authors:** Weiting Liu, Binpeng Zhan, Chunxin Gu, Ping Yu, Guoshi Zhang, Xin Fu, Christian Cipriani, Liang Hu

**Affiliations:** 1State Key Laboratory of Fluid Power and Mechatronic Systems, School of Mechanical Engineering, Zhejiang University, Hangzhou 310007, China; liuwt@zju.edu.cn (W.L.); 11825032@zju.edu.cn (B.Z.); cxgu@zju.edu.cn (C.G.); gszhang@zju.edu.cn (G.Z.); xfu@zju.edu.cn (X.F.); 2College of Mechanical and Electrical Engineering, Wenzhou University, Wenzhou 325035, China; yuping55@wzu.edu.cn; 3The Biorobotic Institute, Scuola Universitaria Superiore Pisa, 56025 Pisa, Italy; christian.cipriani@santannapisa.it

**Keywords:** sparse tactile sensor array, machine learning, neural network, discrimination of curvature, compliant contact

## Abstract

Object curvature plays an important role in grasping and manipulation. To be more exact, local curvature is a more useful information for grasping practically. Vision and touch are the two main methods to extract surface curvature of an object, but vision is often limited since the complete contact area is invisible during manipulation. In this paper, the authors propose an object curvature estimation method based on an artificial neural network algorithm through a lab-developed sparse tactile sensor array. The compliant layer covering on the sensor is indispensable for fitting the curved surface. Three types (plane, convex sphere, and convex cylinder) of sample and each type of sample including 30 different radiuses (1 mm to 30 mm) were used in the experiment. The overall classification accuracy was 93.1%. The average curvature radius estimating error based on an artificial neural network (ANN) algorithm was 1.87 mm. When the radius of curvature was bigger than 5 mm, the average relative error was smaller than 20%. As a comparison, the sensor array density we used in this paper was less than 9/cm^2^, which was smaller than the density of human SAII receptors, but the discrimination result was close to the SAII receptors. Comparison with the curvature discrimination ability of the human body showed that this method has a promising application prospect.

## 1. Introduction

Object curvature is important information that human hands need to acquire when they are engaged in tactile perception [1] and also plays an important role on the contact state [2]. Likewise, it provides reference information for an intelligent hand to adjust its grasping strategy during manipulation. Generally, it is not necessary to know the exact contour of an object going to be manipulated, so local curvature is more useful information for grasping. Currently, vision and touch are the two main ways to deal with the extraction of object surface curvature. In many scenarios, object curvature information can be directly obtained through visual perception [3,4,5]. However, in the case of poor light or during grasping, the contact area between the finger and the object is partly or completely invisible to the vision system. In this case, the local curvature in the contact area can only be acquired through tactile perception. Furthermore, it is noteworthy that the curvature is changeable during grasping when the contact area is non-rigid. Therefore, the discrimination of object curvature must be practically involved along with tactile perception. 

Existing research using only tactile technology to estimate object curvature is limited. In 2011, N. Wettels et al. [6] classified four objects with different curvatures (plane, radius of curvature 10 mm, 3.6 mm, and 1 mm). S. Salehi et al. [7] classified the curvature of three objects (plane, blade, and radius of curvature 2.5 mm). However, the above studies on object curvature discrimination were mainly to classify a limited number of objects with known curvature, and could not provide the specific value of the curvature. In 1991, R. S. Fearing et al. [8,9] tried to use a stress/strain distribution model of a flexible skin layer in the contact state to inversely solve the surface curvature based on Hertz contact theory. In 2017, Y. Kim et al. [10] used the flexible surface of the sensor to fit the object, and used the Light Emitting Diode (LED) and Complementary Metal Oxide Semiconductor (CMOS) image sensor to reconstruct the surface contour of the sensor to measure the curvature. There are some based on other methods such as the 2016 Ran Xu et al. [11] study using fiber Bragg grating (FBG) sensors to measure the curvature, but was only limited to complete tubular and cylindrical objects.

A more common method of curvature discrimination research is that the manipulator moves along the contour of the object, and the surface curvature is solved by the spatial trajectory information of the movement [12,13,14,15,16,17,18]. Back in 1996, M. Charlebois et al. [12,13] planned the exploratory movement of a mechanical tentacle on the surface through contact perception. H. Zhang et al. [14] estimated the surface curvature based on the least square estimation (LSQ) method through rolling contact between the tactile sensor and objects. In 2007, J. Tian et al. [16] rebuilt the surface by tracking along three concurrent curves on the surface to obtain curvature information. In 2017, Ian Abraham et al. [18] used a low-resolution binary contact sensor for ergodic exploration and successfully obtained curvature information. 

To sum up, research on tactile curvature discrimination has not been systematically and extensively investigated. The contour tracking method can detect both convex and concave surfaces, but it has a trivial effect for relatively small objects. Moreover, it takes a lot of time to track the surface contour, so the curvature information cannot be obtained in real time. Therefore, a universal and real-time curvature estimation method using only tactile information is indeed necessary.

Discrimination ability with high spatial resolution like human hands is what people have always desired. For the human hand, there is a mechanoreceptor density for tactile perception of up to 17,000 units in a hand, and the spatial resolution is up to 1.6 mm [19]. Reducing the size of the sensor and increasing the density of sensor units are the most traditional methods [20,21], but is still far from reaching the level of the sensor density of human hands. Moreover, the high-density sensor array is difficult to integrate, the circuit is tedious, and the signal processing is complicated. Therefore, how to use a sparse sensor array to realize high spatial resolution detection is particularly important [22]. Utilizing the compliant layer to couple the contact information, it is possible to discriminate the curvature by means of the tactile sensor array alone [23]. However, the contact process between the object surface and the compliant layer is non-linear. To this end, we used an artificial neural network (ANN) mapping algorithm. It is suitable for solving complicated nonlinear problems with large samples, which has already been widely used in speech recognition, image recognition, and other fields [24].

In this paper, the authors used a sparse tactile sensor array covered with a continuous compliant human-like skin layer [25] to acquire haptical information. An ANN-based curvature discrimination method is put forward. Through the supervised learning of the experimental sample data, a curvature value estimation model was established to extract the object curvature information by tactile perception. It is more meaningful to use a sparse tactile sensor array instead of a high-density sensor array, demonstrating the potential capability of prosthetic applications. Most of the existing tactile curvature discrimination methods are non-real-time or can only classify a limited number of objects, whereas the proposed method here was used to estimate the specific curvature value of the grasped object. Furthermore, the authors extended the discrimination object from a simple spherical object to a cylindrical object. As the relevant research is relatively rare, the authors evaluated the results with the curvature discrimination ability of the human body both in recognizable range and discrimination accuracy. 

The rest of this paper is organized as follows. The tactile sensor device is presented in Section 2, followed by the introduction of the object curvature discrimination methodology based on ANN in Section 3. The detailed experiment and experimental results are presented in Section 4. Subsequently, the results are commented in Section 5. Finally, the conclusion and prospect are discussed in Section 6.

## 2. Tactile Sensor Device

In many studies, the tactile sensor was intentionally added with a compliant layer similar to the skin. Youngwoo Kim et al. designed a fluid-type tactile sensor able to measure the size and depth of heterogeneous substances in elastic bodies that consisted of an image sensor, LED lights, and a touch pad filled with translucent water [10]. The University of Genoa in Italy developed a tactile sensor using piezoelectric conductive rubber as a sensitive material, which consists of 8 × 8 arrays of 64 tactile units. In this research, a thin elastic film was used to cover the sensor as a protective layer [26]. Regarding the flexible tactile sensor, there is a contradiction between the necessity of compliant skin layers and their negative effect on the precise detection of contact pressure [27]. On one hand, the compliant skin layer is indispensable for tactile sensors, which ensures smooth and stable contact with objects and protects the sensitive element from the direct impact of the touched object. Unfortunately, on the other hand, the compliant skin layer brings forward an obvious attenuation effect on the contact pressure signal, deteriorating the effectiveness of contact pressure detection. Furthermore, even if a rigid sensor array is covered by a compliant protective layer, the fragile sensitive element (silicon-based sensing element especially) still cannot be fully protected because it is still in contact with the solid medium, which risks breaking the embedded sensitive silicon diaphragm and weak connecting wires.

In this paper, a lab-developed solid–liquid hybrid sparse tactile sensor array was used [25,28]. The structure of the sensor we used was a compliant layer encapsulating the fluid and the silicon sensing element inside. The authors used fluid to replace some parts of the traditional single soft layer. Unlike traditional hydraulic devices, the fluid chamber of this sensor cell contains the soft part, which also deforms under the fluid pressure. While the compliant layer deforms under external contact to compress the fluid inside, the induced fluid pressure is also reversely loaded onto the compliant layer and makes it deform. Thus, there is a coupling effect between the compliant layer deformation and the encapsulated fluid compression. Therefore, fluid compression can map the deformation state of the compliant layer and directly correspond to the output of the sensing element. The encapsulated sensor cell has good sensing performances with sensitivity of 19.9 mV/N, linearity of 0.998, repeatability error of 3.41%, and hysteresis error of 3.34%. The force sensing range is from 5 mN to 1.6 N.

The mechanical structure explosion diagram is shown in Figure 1a. Some modifications have been made on the part of the flexible layer. The flexible layer changes to a continuous planar compliant layer, as shown in Figure 1b. The sensor array mainly consisted of five parts: rigid base, silicon pressure sensing die, silicone oil liquid layer, polydimethylsiloxane (PDMS) compliant layer, and stainless steel clamping plate. The overall size was 14 mm × 14 mm × 8 mm including nine sensor units in a 3 × 3 layout, with the spacing of sensor units of 4.5 mm. The thickness of the PDMS compliant layer was 5 mm. The selected silicon pressure sensing element was a cubic silicon pressure sensing die (Silicon Microstructures, Inc., SM5108C-060, Milpitas, CA, USA). The silicone oil liquid layer was filled with 50 cSt silicone oil (Dow Corning Inc. pmx-200, Midland, MI, USA). The silicone oil not only acts as a transferring medium of pressure conduction, it can also effectively protect the silicon-based sensor unit. The practical diagram of the sensor array is shown in Figure 1c.

## 3. Methodology

### 3.1. Curvature

As shown in Figure 2, for a point on a differentiable surface in three-dimensional Euclidean space, there exists a unique unit normal vector. Innumerable planes containing the unit normal vector can be made, and these planes are called normal planes. The intersection of any normal plane and surface will form a curve passing through the point, and the curve will have a curvature at the point. The two curvatures obtained from any two normal planes perpendicular to each other are called orthogonal curvatures. When a set of orthogonal curvatures are the maximum and minimum values of such kind of curvatures, respectively, the set of orthogonal curvatures is called the principal curvature, represented by *k*_1_ and *k*_2_. The principal curvature is unique for any point on the surface. Differentiable surfaces can be divided into six different types, according to the positive and negative relationship of principal curvature, as shown in Table 1. The surface of objects often has a variety of surface types, and local small areas can be approximately considered as a single type.

This paper focused on the local curvature at the point of contact between the object and the finger. Considering the compliant tactile sensor used in this paper with a flat surface skin layer, it could not effectively reach unseparated contact with concave ellipsoid, concave cylinder, and hyperboloid. Thus, this paper did not involve the contact of these three types of objects; only plane, convex ellipsoid, and convex cylinder were chosen for validation of the proposed method. Due to the complexity of the unequal principal curvature combination for convex ellipsoid contour perception, even humans can acquire only a simple impression of bending degree from the hand perception instead of having information from whole different direction curvatures. Therefore, this paper focused on a simplified situation where the principal curvature was equal, the so called convex sphere, which can be identified by a unique curvature value.

### 3.2. Basic Theory

The intuitive reason why the tactile sensor array can recognize the curvature of the object surface is that the compliant skin layer above the tactile sensor array can be deformed along with the object shape during contact pressure evolution. Additionally, the deformation will bring the stress change inside the compliant layer, which affects the signal outputs of the underlying tactile sensor array. The surface of the object with different curvature will produce different deformation to the coverage layer, resulting in different signal outputs of the tactile sensor array. There exists an intrinsic mapping relationship between curvature and signal outputs. Based on the theory of Hertz contact (no adhesive contact), this subsection utilizes the contact model between rigid sphere and elastic semi-space to analyze the relationship between the stress distribution in elastic semi-space and the curvature radius of the sphere.

As shown in Figure 3, a rigid sphere with radius *R* comes into Hertz contact with the elastic semi-space surface. The force applied on the hemisphere is *F*, the maximum depth of pressing into the elastic semi-space surface is *d*, and the radius of the contact area is *a*. 

The cylindrical coordinate system is established with the initial contact point position as the origin point *O*. The relationship between stress, radius of curvature, and depth of pressing is shown in Equation (1).
(1)$$\sigma_{z}=-\frac{2E^{*}}{\pi }\cdot \frac{\sqrt{R/d}}{R/d+{\left(z/d\right)}^{2}}$$
where $E^{*}=E/\left(1-v^{2}\right)$; *E* is Young’s modulus of the elastic semi-space material; v is the Poisson’s ratio of the elastic semi-space material; and *z* represents the depth in the z direction. According to Equation (1), as *z*^2^/*d* is generally much larger than *d*, when the depth of pressing and the depth of the sensor array are determined, there is a non-linear mapping relationship between *z* normal stress and the curvature radius *R* of the object. In Figure 4, the relationship curve between *z* normal stress and curvature radius at the point where the depth is *z = 5d* is shown. 

### 3.3. Artificial Neural Network Method

In this paper, a BP (back propagation) neural network was chosen to investigate the curvature discrimination of objects. When the compliant skin layer of the tactile sensor array is in contact with the object, it can fuse the surface topography information of the object and transmit signals in a timely manner through a limited number of tactile sensor arrays. Such information processing is often non-linear. The BP neural network can effectively analyze the non-linear information and is suitable for the investigation of object curvature discrimination.

As mentioned in Section 3.1, the curvature discrimination of contact objects in this paper only considered a plane, convex cylinder, and convex sphere, which also have a significant difference in contact state characteristics. As shown in Figure 5, the contact surface between the convex cylinder and the compliant skin layer was axisymmetric, while the contact surface between the plane and the convex sphere and the compliant skin layer was centrally symmetric. However, the contact surface of the convex sphere will increase with the increase in the depth of pressing, while the contact surface of the plane will not change. Therefore, the BP neural network classification model needs to be established first to classify the surface shape features of unknown objects and determine which surface type it belongs to. Then, corresponding BP neural network curvature radius prediction models should be built for the convex cylinder and convex sphere, respectively, to estimate the specific curvature radius of the contact surface.

As shown in Figure 6, the surface type classification neural network structure and curvature radius estimation neural network structure were established, both of which are BP neural networks with a single hidden layer. The input signal comes from the nine sensing units of the tactile sensor array, each of which contains two types of signal information: time-domain signal information (sensitivity variation) and frequency-domain signal information (principal frequency component), so there are 18 input signal information in total. Considering that curvature value estimation is more complex than classification, the number of units in the hidden layer was set as 30 for surface type classification and 50 for curvature estimation. The output is the code of the three types of surface and a single value of curvature radius, respectively.

After the establishment of the neural network topology structure, the above two neural networks were trained according to the experimental sample data. The samples were randomly divided into 70%, 15%, and 15% for neural network training, verification, and testing, respectively. The validity of the established neural network is analyzed in next section, according to the test results.

## 4. Experiment

### 4.1. Experiment Platform

The experimental loading platform for object curvature discrimination is shown in Figure 7. The three-axis motion stage adopted the M460P of Newport, USA. The maximum stroke in the three directions was 25 mm, and the displacement precision could reach 0.01 mm. Experimental samples with different shapes and curvature radius are shown in Figure 8. In order to ensure the smooth contact process between the object and the tactile sensor array, the loading depth was set as 1 mm (starting from the contact surface), and the loading speed was selected as a constant 0.2 mm/s. The samples were made by 3D printing technology using photosensitive resin, which can be considered as nearly rigid compared with the compliant skin layer of the tactile sensor array. The shape of the samples included three types: plane, convex sphere, and convex cylinder. There were 30 samples with different curvature radius values on the convex sphere and convex cylinder, respectively. The curvature radius range was 1–30 mm, and the difference of curvature radius between adjacent samples was 1 mm.

### 4.2. Experiment Detail

The discrete distribution of sensor elements, the boundary constraint of sensor array, and the limitation of manufacturing process will surely cause the non-uniformity of the sensor array performance, thus the contact position will surely affect the signal combination of the tactile sensor array. For this reason, contact loading at different positions on the surface of the compliant skin layer of the tactile sensor array was carried out, and the acquired sample data at different contact positions was used for neural network training. Obviously, the axis direction of the convex cylinder will also affect the signal output of the sensor array. In order to simplify the analysis, but not lose the generality of validation of the model, the direction parallel to the two axes is applied during convex cylinder experiment. The loading position distribution of the plane and convex sphere for the automatic contact experiment is shown in Figure 9a, and the loading position distribution of convex cylinder is shown in Figure 9b. In order to ensure complete and effective contact between each sample and the compliant skin layer, and considering the surface size of the compliant skin layer, the loading points for the planar or convex spherical samples were set as a 5 × 5 lattice evenly distributed around the center point of the surface origin, with the spacing of the loading points being 1 mm, with 25 different loading positions in total. The loading position for the convex cylindrical sample was set as a 1 × 5 linear array with respect to the X-axis or Y-axis symmetry position, and the loading position interval was 1 mm, with a total of 10 different loading positions.

The raw output data of the sensor array is shown in Figure 10, and sensors 1–9 represent the nine sensing elements in Figure 1. The sample used in Figure 10a is a sphere with a curvature radius of 5 mm, and the loading position is shown in the lower right corner in Figure 9a; the sample used in Figure 10b is a sphere with a curvature radius of 15 mm, and the loading position is the center of Figure 9a. Two kinds of data, named as time-domain data and frequency-domain data, were used as the sample data of the neural network, which were extracted from the signals acquired from the automatic contact experiment. Time-domain data are the measure of the tactile sensor array response sensitivity in a process of stable contact. Since the object loading process is uniform loading at a constant speed, the variation of the output amplitude of the sensor array signal in the same interval is extracted as the steady-state output indicators. Frequency-domain data are the measure of how fast the sensor array responds to the stress change of contact interface, which can be reflected by the change of the principal frequency in the frequency domain. Therefore, fast Fourier transform (FFT) was carried out on the output response signal of the sensor array in the whole contact process to find the corresponding principal frequency as the dynamic response indicators. The sensor array output signal loaded at each position of each experimental sample was recorded, and the data of the above two aspects were also extracted and normalized. The experiments were carried out for the different shapes and different curvature samples shown in Figure 8, and a number of sample data could be extracted for each. Each set of data for the specific sample contained 18 input data (nine time-domain characteristics and nine frequency-domain characteristics of the data) and two output data (object surface type code and radius of curvature, while the surface is plane so only one type of code is needed). Thereby, the corresponding sample database was established for the training and testing of the neural network. During the training, the Levenberg–Marquardt training function was used to adjust the connection weights and threshold of the neural network, and mean square deviation was used to evaluate the output error of the network.

### 4.3. Experimental Results

First, the surface types of the samples were classified, and 70% sample data were randomly selected from the sample database for neural network training. That is, the total amount of data in the sample database was 16,010, of which 11,206 random sample data were used for training. The sample data included surface types, different loading positions, and different curvatures. The confusion matrix results of training, verification, and testing of the neural network for surface type classification are shown in Figure 11, where codes 1, 2 and 3 represent plane, convex sphere, and convex cylinder, respectively. The target group represents the real surface type code, and the output group represents the predicted surface type code. The green box in the figure represents the number and proportion of samples whose predicted code was consistent with the actual code, while the pink box represents the number and proportion of samples whose actual code was incorrectly predicted to be another code. The green number in the light gray box at the bottom indicates the correct prediction proportion in the sample with the actual code, while the red number indicates the wrong prediction proportion. The green number in the light gray box on the right indicates the correct prediction proportion in the sample with corresponding prediction output code, while the red number represents the wrong prediction proportion. The green number in the dark grey box at the bottom right represents the proportion of all correctly predicted samples in the total sample, while the red number represents the proportion of incorrectly predicted samples. 

The experimental results show that the classification accuracy rate of training process was 93.3%, the accuracy of validation process was 91.8%, and the accuracy of testing process was 93.3%. The overall accuracy was 93.1%, considering the different loading positions and rich curvature radius values. The accuracy of this method is similar to that in the existing research, but the existing research does not include any experiments on load position variation, and had less different curvature values used in its experiment [6].

After the object surface type classification, we needed to estimate the curvature radius of the object. Two neural network models, A and B, were built for training according to the sample database of the convex sphere and convex cylinder, respectively. The architecture of model A and model B is shown in Figure 6b. At the same time, a synthetic neural network model C was also established to estimate curvature radius for the hybrid sample database of the convex sphere and convex cylinder in the case of unknown surface types in advance. The synthetic model C is actually a combination of the classification model, model A and model B. The curvature type of the sample is first classified by the classification model, and then the specific curvature value is estimated by the corresponding model A or B. The output of the synthetic model C includes the type of curvature and the specific curvature value. If the type is the plane, then it is output directly. 

The experimental results are shown in Figure 12 and Figure 13. Figure 12 shows the distribution of the average curvature radius estimating error. The error of model A for the convex sphere curvature estimating was 1.18 mm, and the estimating error of 90.4% samples was less than 3 mm. The average curvature radius estimating error of model B for the convex cylinder curvature estimating was 2.75 mm, and the estimating error of 69.7% samples was less than 3 mm. The average curvature radius estimating error of synthetic model C for the mixed convex sphere and cylinder curvature estimating was 1.87 mm and the estimating error of 81.8% samples in total was less than 3 mm. Figure 13 shows the distribution of the mean relative error at different curvature radius. Figure 13a is the curvature radius estimation results of model A for the sphere. When the curvature radius was larger than 5 mm, the relative error was less than 17%. The relative error increased to 60% when the curvature radius was 1 mm (i.e., the mean error was 0.6 mm). Figure 13b is the curvature radius estimated by model B for the cylinder. When the radius of curvature was larger than 5 mm, the relative error was less than 40%, and when the radius of curvature was larger than 7 mm, the relative error was less than 24%. The relative error increased sharply when the curvature radius was close to 1 mm, reaching 300% (i.e., the mean error was 3 mm). Figure 13c is the estimation results of the curvature radius of all types by the synthetic model C. When the curvature radius was larger than 5 mm, the relative error was less than 20%. The relative error increased to 200% when the curvature radius was 1 mm, (i.e., the mean error was 2 mm). Therefore, the established neural network curvature radius estimating model could estimate the surface curvature radius of convex sphere and convex cylinder, respectively, and the synthetic model C could successfully estimate the object curvature radius without the prior information of surface types. 

The model training time was about 10 min with a workstation Intel Xeon E5-2687W v3. The average estimation time was 72 μs for the trained neural network, and the algorithm ran on a PC with an Intel Core i5-4200M processor, 8GB of RAM, and a 64-bit Windows 7 Ultimate operating system.

## 5. Discussion

In this section, we evaluate the performance of the curvature discrimination algorithm based on the sparse sensor array, and compare the results with existing research and the curvature discrimination ability of human fingers. First, the authors discussed the curvature type classification performance of the algorithm. In the existing research on classification of curvature type using tactile technology, N. Wettels et al. [6] classified four samples with an accuracy of 94.7% and S. Salehi et al. [7] classified three samples with an accuracy of 97.5%. In this paper, the authors tested three types of samples and each type included 30 different radius samples (except for the plane). This is equivalent to classifying 61 different samples into three types with an accuracy of 93.1%. Under a more systematic experiment, the accuracy remained at the same level, showing the good performance of the sensor we designed as well as the algorithm. 

More strikingly, the authors estimated the curvature value of the sample. As the relevant research is relatively superficial, we evaluated the results with the curvature discrimination ability of the human body. In 1991, A.W. Goodwin et al. [29] pointed out that fingers can distinguish the difference in curvature between 6.95 mm and 6.33 mm, and the difference was 3.48 mm and 3.13 mm, respectively. About 10% of the curvature difference could be discriminated. A few months later, another experiment [30] showed that when the contact area was constant, a human finger could identify a 13% difference in curvature at 3.50 mm and 18% difference in curvature at 6.49 mm. In 2014, Gregory J. Gerling et al. [31] came to a conclusion similar to Goodwin’s through modeling and simulation. It is worth noting that the aforementioned human finger’s discrimination of curvature does not identify the specific value of curvature, but only determines whether the two curvatures of the given samples are identical. There is still a gap between this and judging the curvature value directly. Our curvature discrimination results showed that when the radius of curvature was bigger than 5 mm, the average relative error was less than 20%. When the radius of the curvature increased, the average relative error decreased. When the radius of curvature was small, the error was indeed large, and the error increased sharply when the radius of curvature was 1 mm. The discrimination effect needs to be improved when the radius of curvature is small, and it may work to improve the discrimination accuracy through separate modeling training.

In more detail, the density of the sensing elements can be evaluated with human hand mechanoreceptor density. There are four types of functional classes of tactile afferents [32]: slowly adapting type I (SAIs); slowly adapting type II (SAIIs); fast-adapting type I (FAIs); and fast-adapting type II (FAIIs). The response of SAIs increased from the curvature radius of 1.92 mm to the plane; the response of SAIIs increased when the curvature radius from 5.81 mm to 12.4 mm, but did not change significantly in other ranges. The other two types (FAIs and FAIIs) had no obvious response to curvature. SAI was mainly distributed at the fingertips, with a density of 70/cm^2^. SAII, on the other hand, was more evenly distributed in the palm, with a density of about 20/cm^2^ [33]. In 2011, Isabelle I. et al. indicated through a validated computational model [34] that a population must have at least 20 sensors/cm^2^ to maintain response resolution in daily living [35], which is close to the density of SAII receptors. As a comparison, the sensor array density we used in this paper was less than 9/cm^2^, which is smaller than the density of human SAII receptors. The discrimination result was close to that of the SAII receptors, and the sensing range was slightly larger than that of the SAII receptors. Although the discrimination effect was slightly weaker than the performance of the finger tips, it is similar to the discrimination performance of other parts of the human hand, which has a very promising application prospect.

## 6. Conclusions and Outlook

In this paper, a previously published sparse tactile sensor array from our laboratory was used to systematically investigate the performance of tactile sensors in curvature response through an artificial neural network including classification and estimation of the curvature value. Abundant, plentiful samples were used in our experiment including plane, spheres, and cylinders and the curvature radius ranged from 1 mm (pen tip, electrical wire) to 30 mm (apple, door handle, water bottle), covering most everyday items and guiding practical applications. The results showed that the classification performance of curvature type by this method was effective, and the classification accuracy reached 93.1%. Regarding the estimation of the curvature radius value, the results showed that the relative error was less than 20% when the radius of curvature was larger than 5 mm. We consider this to be an acceptable result when compared with a human finger experiment, which shows that the lab-developed sensor has very promising application prospects. 

In future works, the following aspects can be improved and expanded based on the work of this paper: explore the method of non-convex object surface curvature discrimination, consider the actual contact situation after integrating the sensor array into an artificial hand, and extend the object curvature discrimination method to an arbitrary contact state.

## Figures and Tables

**Figure 1 micromachines-11-00583-f001:**
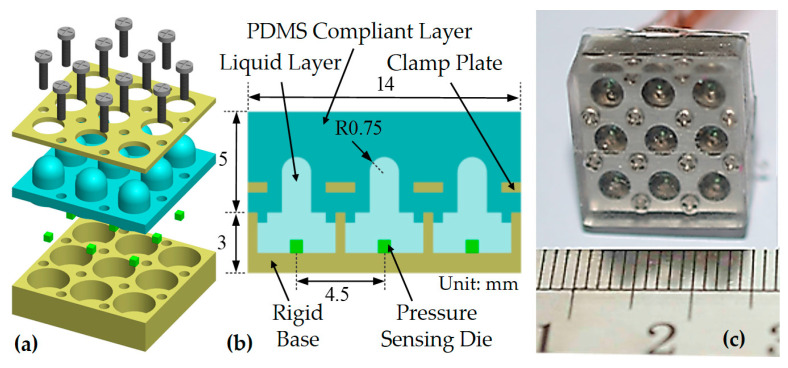
Schematic diagram of the sparse solid–liquid hybrid tactile sensor array structure. (**a**) Explosion diagram of sensor array structure; (**b**) Cross-section diagram of sensor array with continuous planar compliant layer; (**c**) Real prototype of sensor array.

**Figure 2 micromachines-11-00583-f002:**
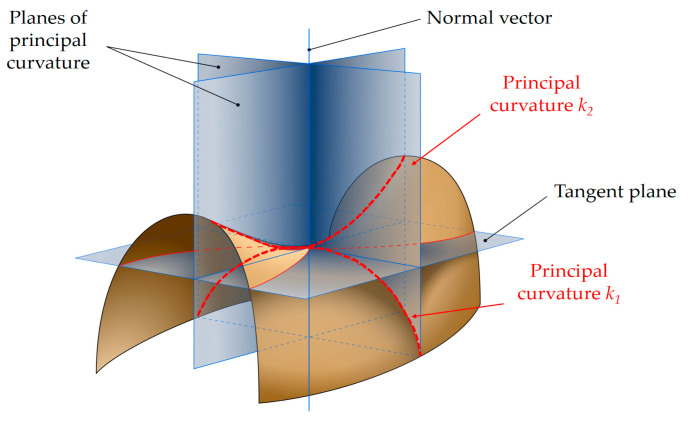
Schematic diagram of the normal plane of the principal curvature at a point on the surface. Source: Wikipedia.

**Figure 3 micromachines-11-00583-f003:**
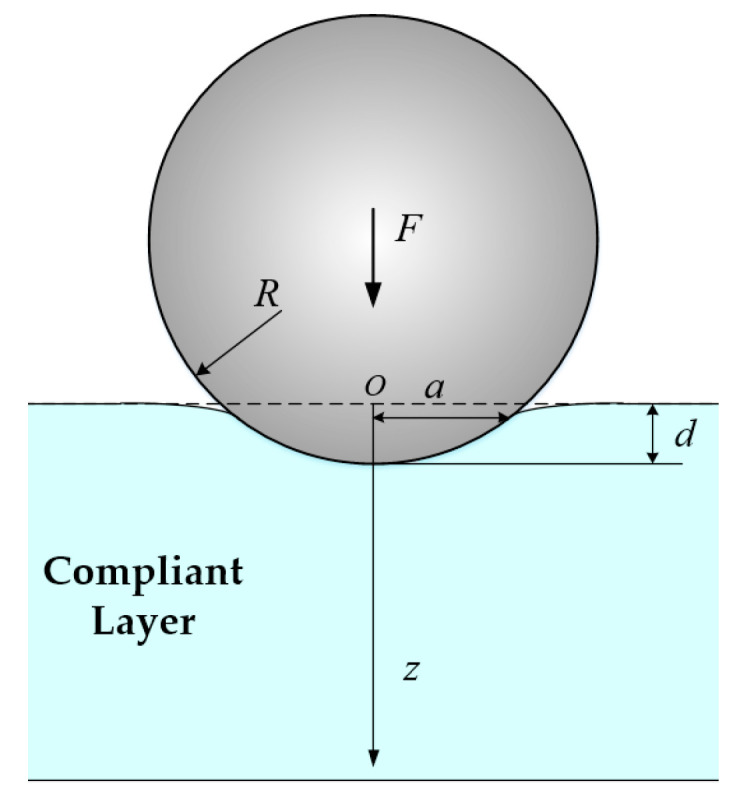
Hertz contact model of rigid sphere and elastic semi-space.

**Figure 4 micromachines-11-00583-f004:**
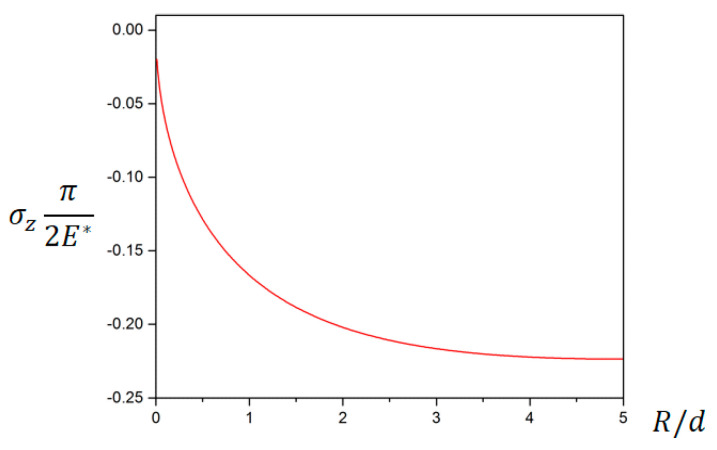
The relationship between *z* normal stress and curvature radius when *z = 5d.*

**Figure 5 micromachines-11-00583-f005:**
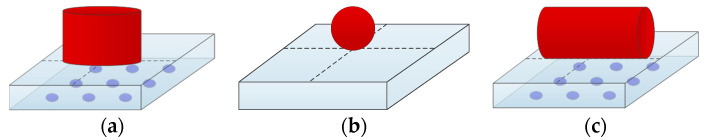
Contact state diagram of the three different surface types: (**a**) plane; (**b**) convex spheres; (**c**) convex cylinder.

**Figure 6 micromachines-11-00583-f006:**
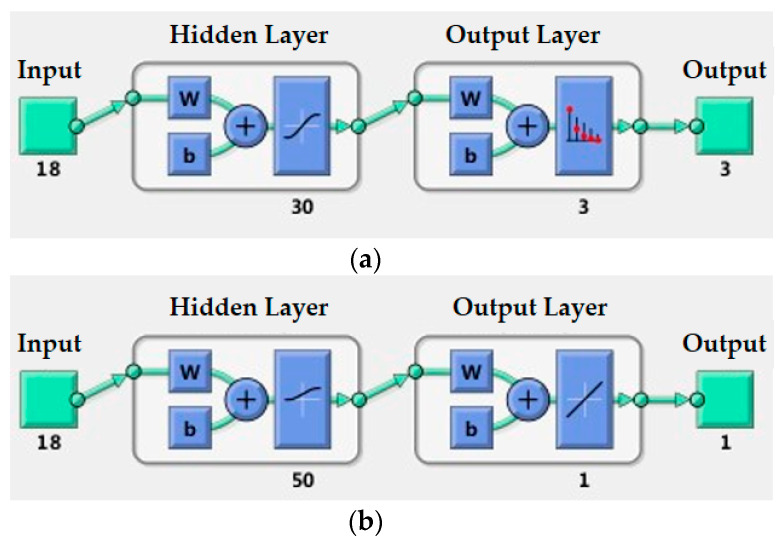
Back propagation (BP) neural network structure for object curvature discrimination: (**a**) single hidden layer surface type classification neural network structure; (**b**) single hidden layer curvature radius estimation neural network structure.

**Figure 7 micromachines-11-00583-f007:**
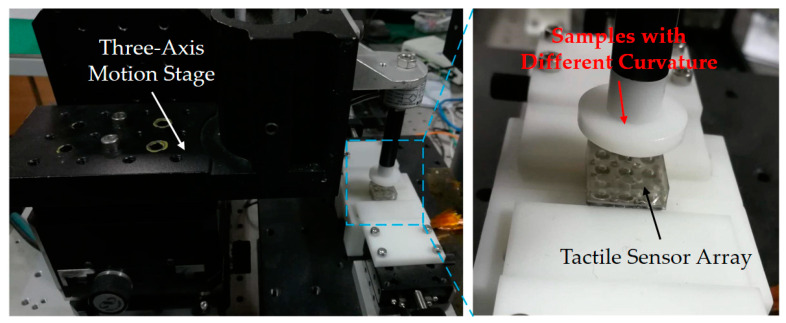
Automatic contact loading test platform for object curvature discrimination.

**Figure 8 micromachines-11-00583-f008:**
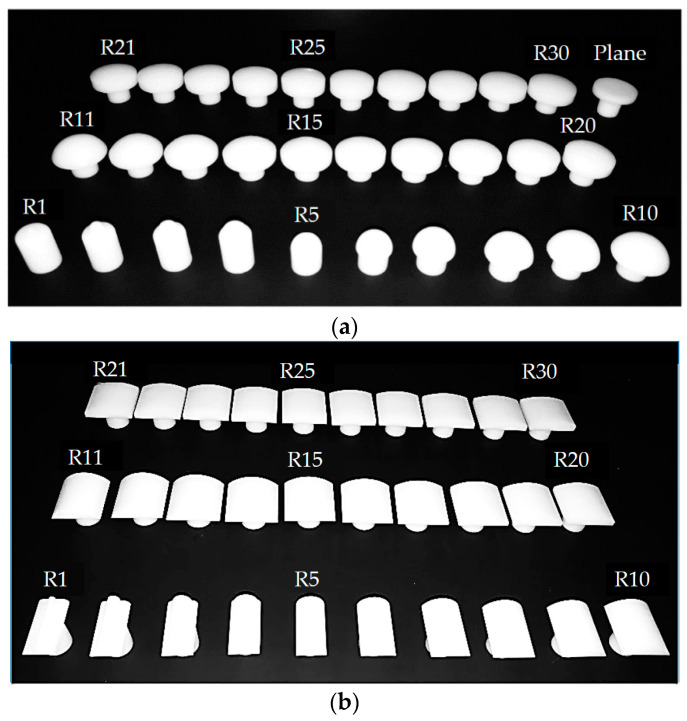
Experimental samples for object curvature discrimination: (**a**) samples of convex sphere and plane objects; (**b**) samples of convex cylindrical objects.

**Figure 9 micromachines-11-00583-f009:**
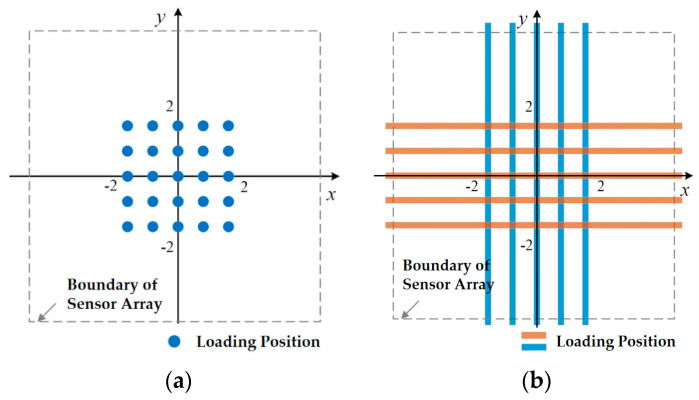
Loading position distribution of automatic object contact experiment: (**a**) loading position of plane and convex sphere; (**b**) loading position of convex cylinder.

**Figure 10 micromachines-11-00583-f010:**
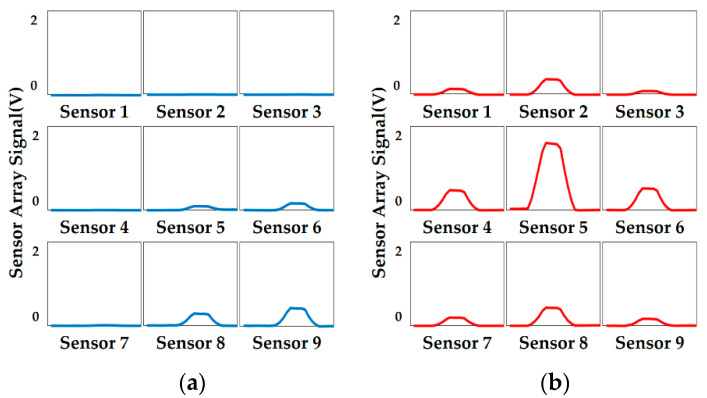
Output data of the sensor array using a sphere sample: (**a**) the curvature radius is 5 mm and the loading position in lower right corner of Figure 9a; (**b**) the curvature radius is 15 mm and the loading position is in the center of Figure 9a.

**Figure 11 micromachines-11-00583-f011:**
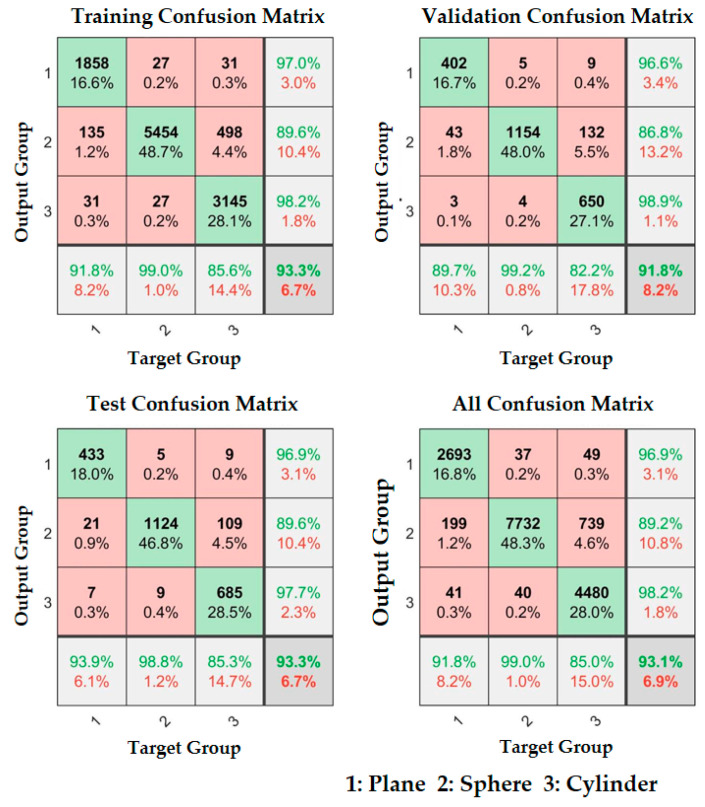
Object surface type classification results.

**Figure 12 micromachines-11-00583-f012:**
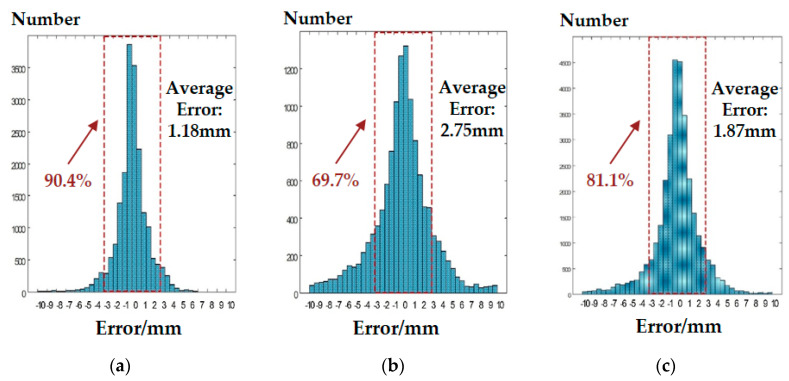
Curvature radius estimating results: (**a**) estimating the error distribution of the convex sphere curvature radius; (**b**) estimating the error distribution of the convex cylinder curvature radius; (**c**) estimating error distribution of the curvature radius with unknown surface types.

**Figure 13 micromachines-11-00583-f013:**
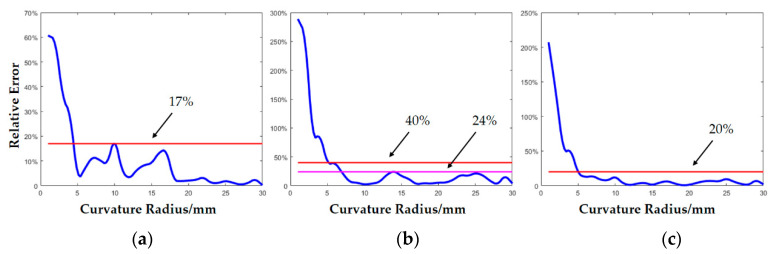
The mean relative error distribution of curvature radius estimation. (**a**) Curvature estimation results of model A for sphere. (**b**) Results of model B for cylinder. (**c**) Results of synthetic model C for all types.

**Table 1 micromachines-11-00583-t001:** Classification of surface types.

Principal Curvature	*k*_1_ < 0	*k*_1_ = 0	*k*_1_ > 0
***k*_2_ < 0**	Concave ellipsoid	Concave cylinder	Hyperboloid
***k*_2_ = 0**	Concave cylinder	Plane	Convex cylinder
***k*_2_ > 0**	Hyperboloid	Convex cylinder	Convex ellipsoid

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
