# Peer review of "Discrimination of Object Curvature Based on a Sparse Tactile Sensor Array"

_micromachines, 2020, doi:10.3390/mi11060583_

Round 1

Reviewer 1 Report

The manuscript proposes a discrimination method of surface curvature. The sensor consists of a contact surface with soft material, liquid, rigid base, and nine pressure sensors. The sensor has a spatial sparse structure. The method discriminated 30 curvatures in an experiment by ANN. The authors concluded that the average relative error for the radius of curvature is bigger than 5 mm was small.
I think the manuscript fits in the scope of the journal.

Comments:

1. What are the specific applications of this sensor and method? What can this sensor do with the experimentally verified accuracy?

2. In section 3.2, the authors shows the relationship between normal stress and curvature in Figure 4. Does the prototype sensor also show the relationship like Figure 4?

3. The authors inputted 18 data (time and frequency domain signal data) to the NN. Please show typical data in the section of the experimental result.

4. In Figure 10, the classification error rates between target group 3 and output group 2 were relatively high in comparison with the others. Why is this? I think the authors should discuss the reason of the difference of the results.

5. In line 347, suddenly, models A and B appeared. Are the models the same with the NN in Figure 6?

Reviewer 2 Report

This paper presents a study on the perception of the object's radius via tactile sensing. The sensor that is used in this research is a self-developed tactile sensor array which measures distributed pressure force signal related to the contact force distribution at the contact patch between object and sensor. The authors then utilize neural networks to classify the contact geometry type as well as the curvature radius. This line of research, as well as the results, looks reasonable.

However, the presentation of this paper should be greatly improved. First, it's too long, with redundancy and with parts that are of little correlation with the main idea.

For example, 1, the introduction is too long and not well summarized. The authors should better organize the related work in the introduction so that it's easier for readers to follow the idea flow.

2, The basic theory about Herzian contact model is quite not related with what the authors are doing. The authors might briefly summarize the idea and intuition behind that, which is enough.

3, The "Discussion" section seems redundant with the introduction. Please consider merging them or discuss more your own work than reviewing others' work at such a section.

4, Please better summarize and be more specific about your contribution in the final "conclusion" section. 

Round 2

Reviewer 2 Report

The revision has addressed my concerns and the manuscript has been dramatically improved. I believe it should be published.

This manuscript is a resubmission of an earlier submission. The following is a list of the peer review reports and author responses from that submission.

Round 1

Reviewer 1 Report

Summary

This manuscript presents an application of a previously published tactile sensor to a curvature discrimination/estimation task. The results are of a fairly low order of accuracy. This work has little novelty in either the tactile sensor or theoretical treatment. The paper itself is not well written, containing numerous grammatical errors and omission of key details. There is a long section discussing the architecture of artificial neural networks which is not relevant and is of a low academic standard.

More detailed comments follow:

Introduction

Literature review of relevant research is limited. Two of the references are of the authors’ previous work which seemed to be of a higher standard. The differences between the current manuscript and these previously published works have not been sufficiently explained.

“Most of the current tactile curvature recognition methods can only classify and identify a limited number of objects, and unable to calculate the specific curvature value. Whereas, the proposed method here is tried to evaluate the specific value of the curvature of an object grasped.” This statement is not substantiated by references and, in fact, the research presented here also involves use of a classifier.

“Furthermore, the authors extend the recognition object from simple spherical object to cylindrical

object.” Actually, a sphere has curvature in 2 dimensions and is therefore more challenging than a cylinder.

Tactile Sensor Device

“Tactile sensor is the core device to acquire tactile information.” This is a tautology.

The principle of transduction of the sensor is not explained. Two previous works by the same authors both present apparently identical devices: if that is the case it should be clearly acknowledged. If this device differs, the differences need to be explained.

Methodology

The theoretical treatment using Hertzian contact model is identical to that presented in reference 8: “A Novel Inverse Solution of Contact Force Based on a Sparse Tactile Sensor Array”.

The description of neural networks is very simplistic and completely unnecessary for such a well-known architecture.

Experimental Method

The setup used is not clearly explained. Figure 9 shows an arrow to something referred to as a “standard force sensor” which is not mentioned in the text.

Results

The accuracy of curvature prediction for the samples with smaller radii of curvature is extremely poor – average errors of ~2mm for radii ranging from 1mm to 30mm. This does not seem to have been commented on in the conclusions.

Conclusions

The first paragraph, and therefore majority of the conclusions are, in fact, just re-stating the introduction and methods.

The main conclusion that the “method can effectively estimate the object curvature radius” has not been demonstrated to an acceptable degree of accuracy.

Novelty

It would appear that the only novelty in the current manuscript is the use of neural networks to classify surfaces as either locally spherical, cylindrical or planar and then to use a second network to predict the local curvature by regression. The details of this second network (referred to as “synthetic model C” are not stated at all. Given this lack of novelty, the current manuscript cannot be recommended for journal publication. It might be more appropriate for a conference paper (with substantial editing).

Reviewer 2 Report

The authors focused on the discrimination of object curvature by the tactile sensor array. The tactile sensor consists of 9 pressure sensor elements and soft material layers. The alignment of the elements is sparse. To discriminate the curvature, the authors used ANN. The experiment validated the discrimination rate by using 60 samples.

The reviewer's comments are as follows:

1. Please show the fundamental characteristics of the tactile sensor. For example, sensitivity, range, hysteresis and so on.

2. Figure 4 shows the relationship between normal stress and curvature radius. What is the relationship between normal stress and pressure measurement value?

3. Figure 4 shows a simple relationship. So, it looks that the authors can calculate the curvature from 9 measurement data. Is it right?

4. How long was the training time? How many training data did you use?

5. Please show the target value of the discrimination rate. Is the target value enough for some applications?

6. Please discuss the generalization performance of the ANN.

7. The reviewer considers the experimental system vertically pressed a sample to the sensor. If it is right, please add a discussion in the case of not vertical press or angled sample.